# Antibiotic—*Lysobacter enzymogenes* proteases combination as a novel virulence attenuating therapy

Ghadeer A. R. Y. Suaifan[1]*, Diana M. A. Abdel Rahman[1], Ala' M. Abu-Odeh[2‡], Fahid Abu Jbara[3‡], Mayadah B. Shehadeh[1], Rula M. Darwish[4]

1 Department of Pharmaceutical Sciences, School of Pharmacy, The University of Jordan, Amman, Jordan,
2 Department of Pharmaceutical Chemistry and Pharmacognosy, School of Pharmacy, Applied Science Private University, Jordan, Amman, 3 School of Medicine, The University of Jordan, Amman, Jordan,
4 Department of Pharmaceutics and Pharmaceutical Biotechnology, School of Pharmacy, The University of Jordan, Amman, Jordan

☉ These authors contributed equally to this work.
‡ AMAO and FAJ also contributed equally to this work.
* gh.suaifan@ju.edu.jo, Ghadeer_petra@yahoo.com

**Data Availability Statement:** All relevant data are within the manuscript and its Supporting information files.

## Abstract

Minimizing antibiotic resistance is a key motivation strategy in designing and developing new and combination therapy. In this study, a combination of the antibiotics (cefixime, levofloxacin and gentamicin) with *Lysobacter enzymogenes* (*L. enzymogenes*) bioactive proteases present in the cell-free supernatant (CFS) have been investigated against the Gram-positive methicillin-sensitive *Staphylococcus aureus* (MSSA), methicillin-resistant *Staphylococcus aureus* (MRSA) and the Gram-negative *Escherichia coli* (*E. coli* O157:H7). Results indicated that *L. enzymogenes* CFS had maximum proteolytic activity after 11 days of incubation and higher growth inhibitory properties against MSSA and MRSA compared to *E. coli* (O157:H7). The combination of *L. enzymogenes* CFS with cefixime, gentamicin and levofloxacin at sub-MIC levels, has potentiated their bacterial inhibition capacity. Interestingly, combining cefixime with *L. enzymogenes* CFS restored its antibacterial activity against MRSA. The MTT assay revealed that *L. enzymogenes* CFS has no significant reduction in human normal skin fibroblast (CCD-1064SK) cell viability. In conclusion, *L. enzymogenes* bioactive proteases are natural potentiators for antimicrobials with different bacterial targets including cefixime, gentamicin and levofloxacin representing the beginning of a modern and efficient era in the battle against multidrug-resistant pathogens.

## Introduction

Infectious diseases are one of the leading causes of illness and death among individuals worldwide [1]. A wide variety of infections can be caused by either primary or opportunistic pathogens [2] which can acquire and transfer resistance genetically by either mutation or gene transfer thus, microbes control became a challenge [3]. The World Health Organization (WHO) has considered the emergence of antibiotic-resistant bacterial strains as one of the

**Funding:** This research was supported by a grant (2073, 2460) from the Deanship of the Scientific Research at The University of Jordan and grant (2017) from Hamdi Mango Center for Scientist Research. The funders had no role in study design, data collection and analysis, decision to publish, or preparation of the manuscript.

**Competing interests:** The authors have declared that no competing interests exist.

three most critical public health threats of the 21st century [4]. The incessant development of new antibiotics with a novel mode of action must be intensified to combat drug-resistant infections.

Most clinically relevant antibiotics are derived from natural products (actinomycetes or fungi), their semisynthetic isolates or synthetic [5]. The mining of these antibiotics commonly resulted in the discovery of similar compounds, suggesting an urgent need for a shift in attention to utilize previously uncharacterized microbes as a source of novel antibiotics. A relevant number of plant-beneficial bacteria are effective biocontrol agents against plant pathogenic microorganisms and are referred to as 'green' biopesticides [6]. These microorganisms played an important role in increasing crop production by protecting plants from infectious pathogens, hence lowering the use of carcinogenic pesticides in agriculture [6, 7]. Within the soil microbiome, the *Lysobacter* genus acquired high attention due to its role in controlling pathogen-induced plant diseases [8]. For example, *Lysobacter enzymogenes* (*L. enzymogenes*) C3 and OH11, were reported to be effective in controlling *Bipolaris* leaf-spot on tall fescue and anthracnose on pear fruit, caused by *Bipolaris sorokiniana* and *Colletotrichum fructicola* fungal pathogens, respectively [9, 10]. Thus, *Lysobacter* species have been designated as facultative predators able to lyse several microorganisms including nematodes, bacteria and fungi [11, 12] through epibiotic predation and cell-cell contact. Worth mentioning that the molecular mechanism involved in *Lysobacter* genus killing behavior, is still not fully explored. However, Xi Shen research group revealed that *L. enzymogenes* OH11 uses Type IV Secretion System (T4SS) as the main contact-dependent weapon against other soil-borne bacteria [13, 14]. Other studies indicated that *L. enzymogenes* bacteriolytic activity is attributed to its extracellular alpha (a serine protease) and beta (metalloprotease) lytic enzymes [15]. These proteases together with other enzymes such as chitinases, glucanases, lipases and phospholipases can degrade the cell wall of some plant pathogens including Gram-positive and Gram-negative bacteria [15–17]. Thereafter, *Lysobacter* has emerged as a new source of bioactive natural products [18, 19] and attention has been pointed to their ability to lyse both prokaryotic and eukaryotic microbes with the production of peptides that damage microorganisms' cell walls or membranes at a very low concentration [20].

While the use of bioactive natural products alone or in combination to combat antibacterial resistance is increasingly applied [21–25], several antimicrobial combinations have been studied for synergy *in vitro* and *in vivo* against Gram-positive and Gram-negative pathogens [19, 25–40]. However, limited research has been carried out to evaluate the effect of combining antibiotics with *L. enzymogenes* bioactive metabolites, as a virulence-attenuating combination therapy against resistant strains.

This study aims to investigate the potential implications of *L. enzymogenes* cell-free supernatant (CFS) bioactive products alone and in combination with cefixime, levofloxacin and gentamicin antibiotics against clinically important pathogens including *staphylococcus aureus* (*S. aureus*), one of the most notorious species causing mild and serious infections such as necrotizing pneumonia, septicemia and bone infections [41], *Escherichia coli* (*E. coli*) O157:H7, a well-described food-borne pathogen which produces virulence factors [42] and is responsible for bloody diarrhea and hemolytic-uremic syndrome (HUS) [43].

## Materials and methods

### Bacterial strains and bacterial broth culture preparation

Microorganisms used in this study included *L. enzymogenes* ATCC 29487, methicillin-sensitive *S. aureus* (MSSA) (ATCC 25923), methicillin-resistance *S. aureus* (MRSA) (ATCC 33591) and *E. coli* O157:H7. Culture media used included brain heart infusion (BHI) agar and broth

obtained from Sigma- Aldrich (India) and nutrient agar (NA) obtained from Scharlau (European Union). Sterile syringe filters (0.22 μm) were purchased from Millipore (Amman, Jordan). Antibiotics (cefixime, gentamicin and levofloxacin) were kindly donated by JOSWE® Medical and Dar Al Dawa® pharmaceutical company, Amman-Jordan.

*L. enzymogenes* bacterial pellets were rehydrated aseptically using 5 mL of 10% strength tryptone soy broth (TSB), incubated at 28˚C and shaken at 200 rpm for 3 days in a shaker incubator (MS, Taiwan) [44]. Then, *L. enzymogenes* culture (200 μL) was seeded over 10% strength of tryptone soy agar (TSA) plate and incubated at 28˚C for 3 days. *L. enzymogenes* pure broth culture (PBC) was prepared by transferring a full loop from the stock plate to a 50 mL sterile falcon containing 10% TSB, followed by incubation at 28˚C for 3 days. *E. coli* O157:H7 was cultured in nutrient broth (NB) (Biolab); MSSA and MRSA in brain heart infusion (BHI) broth (Biolab).

## Sigma's non-specific protease activity assay

Casein from bovine milk, Folin & Ciocalteu's phenol reagent, trichloroacetic acid, potassium phosphate buffer (pH 7.5), and anhydrous sodium acetate were purchased from Sigma-Aldrich (USA). Anhydrous sodium carbonate ($Na_2CO_3$) was purchased from SDFCL (India). L-tyrosine was purchased from Hopkin and Williams Ltd. (England).

## Preparation of *L. enzymogenes* cell-free supernatant (CFS) bioactive products and proteolytic activity assessment

Two different pools of *L. enzymogenes* PBC ($10^7$ CFU/mL) were prepared and were assigned as pool A (PA, 4% (v/v)) and pool B (PB, 8% (v/v)). PA was prepared by mixing 0.5 mL of the PBC in a 15 mL falcon tube containing 12 mL 10% TSB and incubating the mixtures at 28˚C for different time intervals (1, 2, 3, 4, 7, 10, 11, 14 and 18 days) with continuous shaking at 200 rpm. PB was prepared using 1 mL of the PBC following the steps used for PA (S1 Fig). The plate count method was applied to enumerate the incubated PA and PB culture series viable cells. Later, *L. enzymogenes* cells of PA and PB cultures series were collected by centrifugation (15000 rpm at 4˚C for 15 min), filtration using a sterile syringe filter (0.22 μm) to obtain the cell-free supernatant (CFS) and stored at -20˚C in small aliquots for later use. The proteolytic activity of the obtained CFS series (Fig 1) was measured using Sigma's non-specific protease activity assay [45]. In this assay, *L. enzymogenes* proteases digest casein substrate to liberate free tyrosine which reacts with Folin's reagent (Folin and Ciocalteu's) to generate a blue color solution. The absorbance of this solution was measured at 660 nm using a microtiter plate reader (Epoch-Biotec, California, USA) and compared with the absorbance of different standard tyrosine concentrations (S2 Fig). All experiments were performed in triplicate.

*L. enzymogenes* CFS proteolytic activity was correlated with *L. enzymogenes* viable cell count and determined in terms of units/mL, which is corresponding to the micromoles (μmol) of tyrosine released from casein per minute, by applying the following equation [45].

$$\text{Protease activity} \left(\frac{\text{Units}}{\text{mL}}\right)$$

$$= \frac{(\text{Tyrosine equivalent liberated}(\mu mol)) \times (\text{Total volume of the assay} * (11 \ mL))}{\text{Volume of protease } (1 \ mL) \times \text{Time of assay } (10 \ min) \times \text{Volume used in colorimetric determination } (2 \ mL)}$$

$*$ = *L. enzymogenes* CFS (1mL) + Casein solution (5mL) + Trichloroacetic acid (5mL).

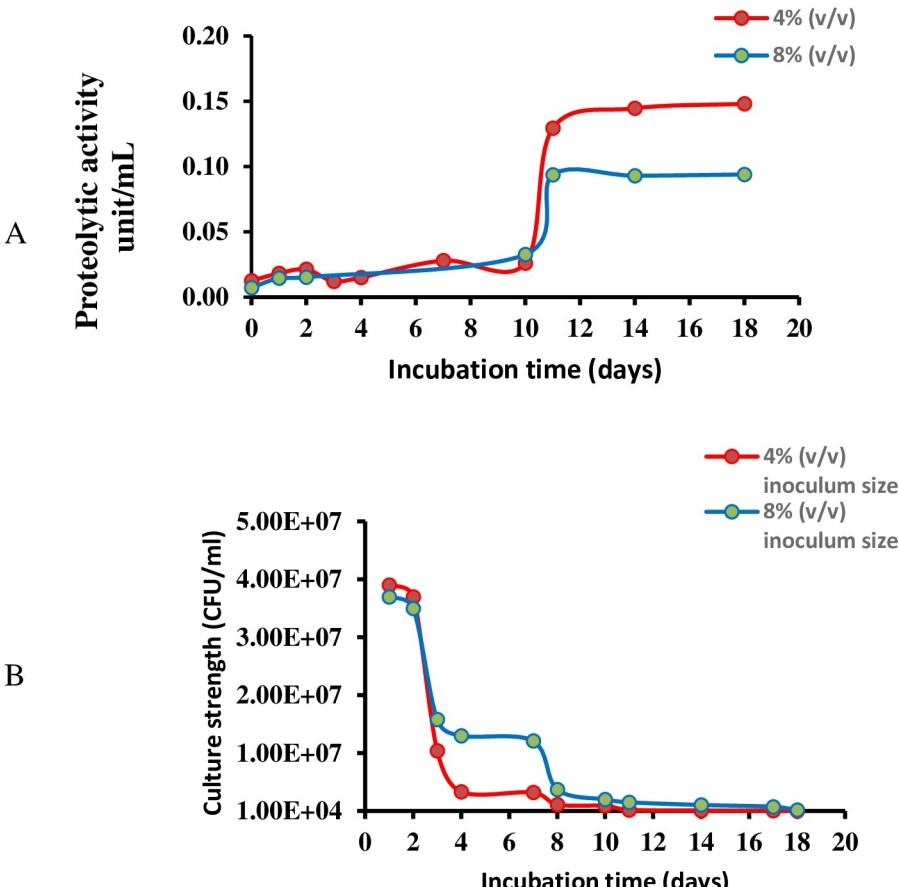

**Fig 1. Effect of incubation time and bacterial density on *L. enzymogenes* proteolytic activity. A**; Effect of culture density (PA, 4% (v/v) and PB, 8% (v/v)) **B**; Effect of incubation time (days).

### Screening of *L. enzymogenes* CFS antibacterial activity

Growth inhibitory activity of *L. enzymogenes* CFS against MSSA, MRSA and *E. coli* O157:H7 PBCs ($10^8$ CFU/mL) grown on agar plates was investigated by spreading two different volumes (100 μL and 200 μL) of *L. enzymogenes* CFS (prepared from a 14-day PBC) over the plates (Fig 2).

### Determination of antibiotics' minimum inhibitory concentrations (MIC)

The susceptibility of MSSA, MRSA and *E. coli* O157:H7 to the three antibiotics; cefixime, levofloxacin and gentamicin was performed using National Committee for Clinical Laboratory Standards (NCCLS) broth microdilution method [46]. Minimum inhibitory concentration (MIC) was determined using 96 flat-bottom microtiter plates (TPP, Switzerland). Each test well was filled with 90 μL BHI for *S. aureus* and NB for *E. coli* O157:H7. An aliquot (100 μL) of the antibiotic stock solution was added to the test well and mixed. A series of twelve 2-fold serial dilutions of the antibiotics were examined. The concentration ranges used to determine MICs were: cefixime 256–0.125 μg/mL, levofloxacin 64–0.031μg/mL, gentamicin 128–0.062 μg/mL against *S. aureus* and cefixime 32–0.0156 μg/mL, levofloxacin 32–0.0156 μg/mL, gentamicin 64–0.031 μg/mL against *E. coli* O157:H7. All dilutions of the tested antibiotics were inoculated with 10 μL of $10^6$ CFU/mL of the specified bacterial strain and then, incubated

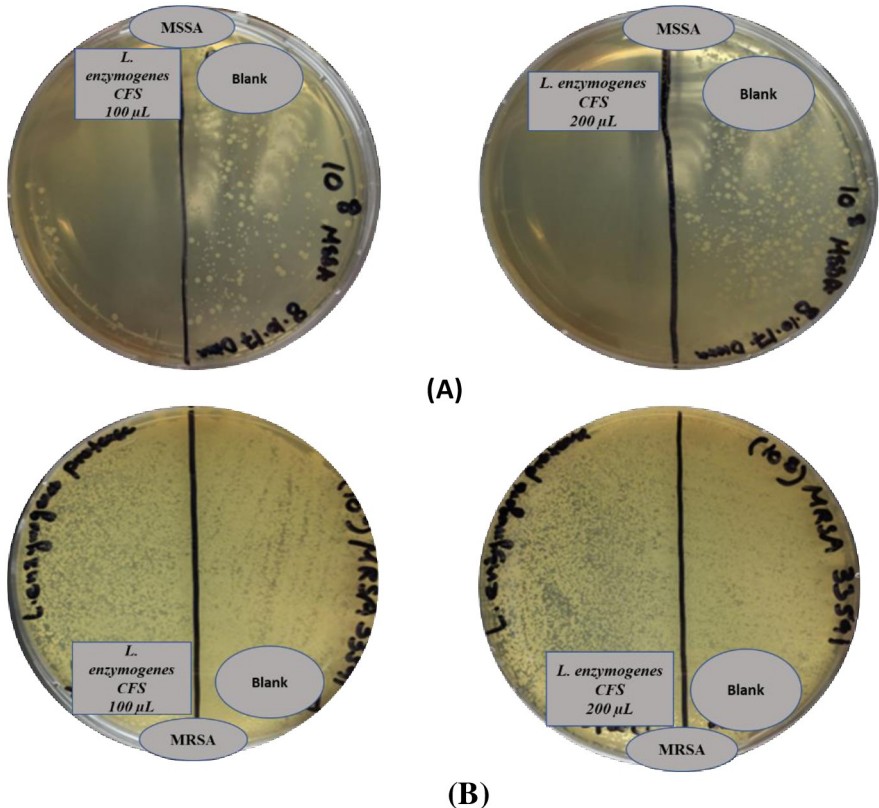

**Fig 2.** *L. enzymogenes* CFS growth inhibitory activity against; (A) MSSA, (B) MRSA using 100 μL and 200 μL *L. enzymogenes* CFS aliquots.

at 37 ˚C for 24 h. Positive control (broth and bacterial suspension) and negative control (broth only) wells were included in every experiment to prove adequate microbial growth and media sterility during the incubation period.

In the test wells, microbial growth was assessed visually from culture turbidity and compared to the negative and positive controls. MICs were determined as the lowest concentration of the antibiotic that inhibits the growth of the microorganism. The test was carried out in triplicate (in the same 96-well plate) and repeated twice for each bacterium.

### Growth inhibition activity of *L. enzymogenes* CFS, antibiotics and their combination

A microplate growth inhibition assay [47] was applied to measure the growth inhibitory effect of *L. enzymogenes* CFS, antibiotics (cefixime, levofloxacin and gentamicin) and their combination against MSSA, MRSA and *E. coli* O157:H7. This assay allows the observation of discernible inhibition during growth using turbidity parameter which is measured through the detection of light scatter in absorbance at 600 nm using a microplate reader.

*L. enzymogenes* CFS growth inhibitory activity evaluation was performed by adding 10 μL aliquots of $10^6$ CFU/mL bacterial suspension (MSSA, MRSA and *E. coli* O157:H7) to sterile microtiter plate wells containing 10 μL *L. enzymogenes* CFS (proteolytic activity 0.145 unit/mL) and 180 μL broth.

For the evaluation of antibiotics' antibacterial activity, the MIC, 0.5 MIC, 0.25 MIC and 0.125 MIC concentrations for each antibiotic (cefixime, levofloxacin and gentamicin) were

prepared. The test was carried out by placing 180 μL BHI for *S. aureus* and NB for *E. coli* O157:H7 and 10 μL of the prepared antibiotic dilution in each well, then 10 μL aliquot of the pathogen ($10^6$ CFU/ml) was added.

For the combination study, 10 μL of the bacterial suspension ($10^6$ CFU/ml) was added to wells containing 170 μL broth, 10 μL of *L. enzymogenes* CFS and 10 μL of each prepared antibiotic concentration (MIC, 0.5 MIC, 0.25 MIC and 0.125 MIC).

As a full growth control run, a 10 μL aliquot of the pathogenic cell suspension was inoculated at $10^6$ CFU/mL into 190 μL of the corresponding sterile broth. Also, a test blank was run with each experiment where the 10 μL of the pathogen was replaced by the proper sterile broth.

In the above-mentioned experiments, the microtiter plate was incubated at 37°C for 24 h and the optical density (OD) was measured at 600 nm using a microplate reader (Epoch-Biotec, California, USA). Results were calculated as the average mean of three readings.

In this study, antibacterial activity was expressed as percentage inhibition of bacterial growth following 24 h incubation at 37 °C and calculated using the following equation:

$$\% \text{ Growth Inhibition} = 100 - \left( \frac{\text{OD}_\text{T} - \text{OD}_\text{TB}}{\text{OD}_\text{FG} - \text{OD}_\text{FB}} \right) \times 100\%$$

Where:

- **$\text{OD}_\text{T}$** is the average optical density of three replicates at 600 nm for the tested target (*L. enzymogenes* CFS, antibiotics or the combination)

- **$\text{OD}_\text{TB}$** is the average optical density of three replicates at 600 nm for the tested target blank (10 μL of the pathogen is replaced by equivolume of the appropriate sterile broth)

- **$\text{OD}_\text{FG}$** is the average optical density of three replicates at 600 nm for the study pathogen full growth.

- **$\text{OD}_\text{FGB}$** is the average optical density of three replicates at 600 nm for the study pathogen blank (No pathogen, only 10 μL of broth was used).

## Cytotoxic activity

MTT assay was applied to evaluate *L. enzymogenes* CFS cytotoxicity. The cell line used in this assay was human normal skin fibroblast (CCD-1064SK) purchased from the American Type Culture Collection (ATCC, Manassas, VA, USA). This colorimetric assay measures the cellular metabolic activity based on the ability of nicotinamide adenine dinucleotide phosphate (NADPH)-dependent cellular oxidoreductase enzymes to reduce the yellow 3-(4,5-dimethylthiazol-2yl)-2,5-diphenyltetrazolium bromide dye (MTT) to form the insoluble formazan purple crystals [48–50]. Cells were seeded in a 96-well culture plates in a final volume of 100 μL media per well, then plates were incubated in a humidified atmosphere (37°C, 5% $CO_2$) for 24 h to allow cells to adhere. When cells reached confluency, they were treated with the *L. enzymogenes* CFS to obtain final concentrations of 1.25, 2.5, 5, 10, 20, 25, 40, and 50% (v/v). Cells were incubated in the humidified atmosphere (37°C, 5% $CO_2$) for 72 h before performing the MTT assay to determine the cells' viability. The assay was conducted in triplicate, and the control cells were treated only with 10% TSB. The optical density for the treated and control wells was measured at 570 nm using a microtiter plate reader (BioTek, Winooski,

VT, USA). Percentage viability was calculated using the formula below

$$\% \text{ Viability} = \left( \frac{\text{Mean OD sample}}{\text{Mean OD blank}} \right) \times 100$$

## Statistical analysis

Data were expressed as a mean ± standard deviation (SD). The statistical significance between different test conditions was determined using the independent sample t-test. The difference among groups was significant when $p < 0.05$.

## Results

### *L. enzymogenes* cell-free supernatant (CFS) proteolytic activity assessment

*L. enzymogenes* CFS proteolytic activity was measured using Sigma's universal protease activity assay. *L. enzymogenes* CFS proteolytic activity was evaluated under two variables; bacterial density (PA, 4% (v/v) and PB, 8% (v/v)) and incubation period (1, 2, 3, 4, 7, 10, 11, 14 and18 days) as presented in Fig 1. Irrespective of the bacterial density (PA and PB), comparable proteolytic activity was observed during the first ten days of incubation (Fig 1A). However, on day eleven, a change in proteolytic activity was detected. The maximal proteolytic activity for the PA and PB stocks were 0.129 unit/mL and 0.093 unit/mL (Fig 1A), respectively. Accordingly, higher inoculum size did not yield higher proteases production.

To gain insight into the characters of bioactive products secreted by *L. enzymogenes*, viable cell count of PA 4% (v/v) and PB 8% (v/v) stocks at different incubation periods was evaluated (Fig 1B). Results revealed that *L. enzymogenes* multiplication process was active during the first two incubation days, thereafter the growth rate decelerates due to depletion of the essential nutrients. Following incubation for eleven days, bacterial death for PA and PB cultures was dominant (Fig 1B) and so it may be proposed that *L. enzymogenes* intracellular proteases were released signifying the maximum proteolytic activity observed (Fig 1A).

### Bacterial inhibition capacity of *L. enzymogenes* CFS against MSSA and MRSA

*L. enzymogenes* CFS growth inhibitory effect was first screened against *S. aureus*. A reduction in MSSA and MRSA colonies density and size was visually noticed when *L. enzymogenes* CFS (100 and 200 μL) was flooded over a plate cultured with $10^8$ CFU/mL of *S. aureus* as shown in Fig 2A and 2B.

### Growth inhibition activity of *L. enzymogenes* CFS, antibiotics and their combination

The growth inhibitory strength of *L. enzymogenes* CFS against different pathogens (MSSA, MRSA and *E. coli* O157:H7) at different concentrations ($10^2$, $10^3$, $10^4$, $10^5$ and $10^6$ CFU/mL) is illustrated in Fig 3. *L. enzymogenes* CFS displayed higher growth inhibitory activity against MSSA and MRSA (16% and 10%, respectively) compared to *E. coli* O157:H7 (5%).

The MICs of the standard antibiotics (cefixime, levofloxacin and gentamicin) on the selected pathogens are summarized in Table 1.

The antibacterial activity of the standard antibiotics (cefixime, gentamicin and levofloxacin) at 0.5 MIC (Fig 4A), 0.25 MIC (Fig 4B) and 0.125 MIC (Fig 4C) alone and in combination with *L. enzymogenes* CFS against MSSA, MRSA and *E coli* O157:H7 are presented in Fig 4. The

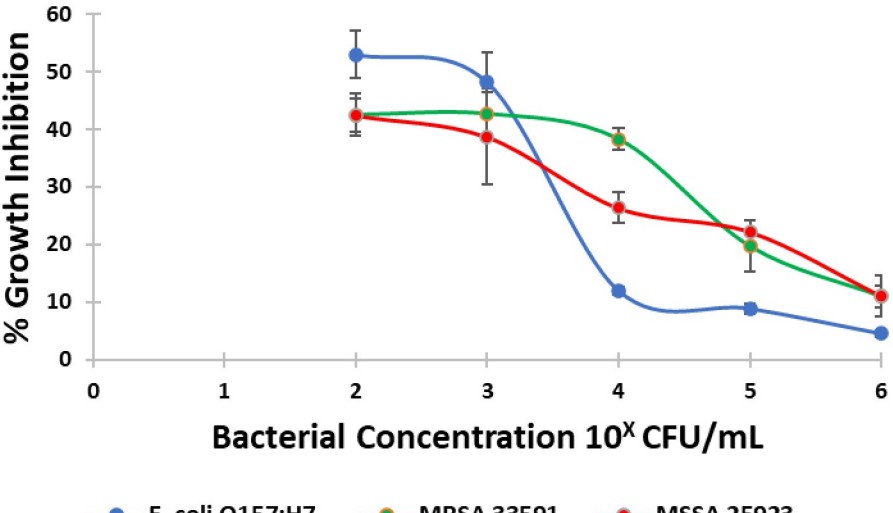

**Fig 3. Growth inhibition activity of *L. enzymogenes* CFS against MSSA, MRSA and *E. coli* O157: H7 at different concentrations ($10^2$, $10^3$, $10^4$, $10^5$ and $10^6$ CFU/mL).**

combination of *L. enzymogenes* CFS with cefixime, gentamicin, levofloxacin at sub-MICs showed an additive effect against MSSA and MRSA (Fig 4). *L. enzymogenes* CFS-antibiotics 0.5 MICs mixtures presented fifty percent or higher growth inhibition activity against MSSA. Apparently, *L. enzymogenes* CFS triggered cefixime's antibacterial activity against MSSA when tested at its 0.5 MIC ($p = 0.001$, Fig 4A), 0.25 MIC ($p = 0.004$, Fig 4B) and 0.125 MIC ($p = 0.045$, Fig 4C). Yet, no significant additive effect for *L. enzymogenes* CFS when combined with cefixime against the Gram-negative *E. coli* O157:H7 at the sub-MIC values. On the contrary, gentamicin and levofloxacin combination at sub-MIC strength with *L. enzymogenes* CFS increased their antibacterial activity against all tested pathogens (Fig 5).

## Cytotoxicity assay

MTT assay results shown in Fig 5 indicated high cell viability values of 92–145% when exposed to different *L. enzymogenes* CFS concentrations (1.25, 2.5, 5, 10, 20, 25, 40, and 50% (v/v)).

## Discussion

Interest in antibiotic adjuvants therapy has increasingly attracted attention within contemporary studies due to the emergence of multidrug-resistant organisms [51–53]. Perhaps one of the leading causes of this resistance is the low microbial cell membrane permeability to antibiotics. Hence, microbial proteases able to perturb other pathogens membrane structure arises as an efficient tool to increase antibiotic bioavailability. Indeed, recent studies reported the

**Table 1. MICs of the antibiotics determined for bacterial strains used in this study.**

| Antibiotics | MSSA | MRSA | *E. coli* O157:H7 |
|---|---|---|---|
| Cefixime | 8 µg/mL | > 32 µg/mL | 0.0312 µg/mL |
| Levofloxacin | 0.25 µg/mL | 0.25 µg/mL | 0.25 µg/mL |
| Gentamicin | 25 µg/mL | 50 µg/mL | 0.375 µg/mL |

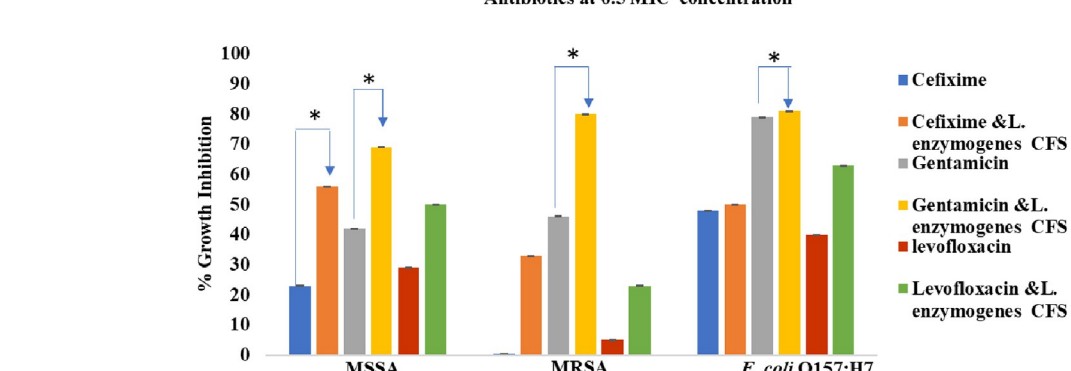

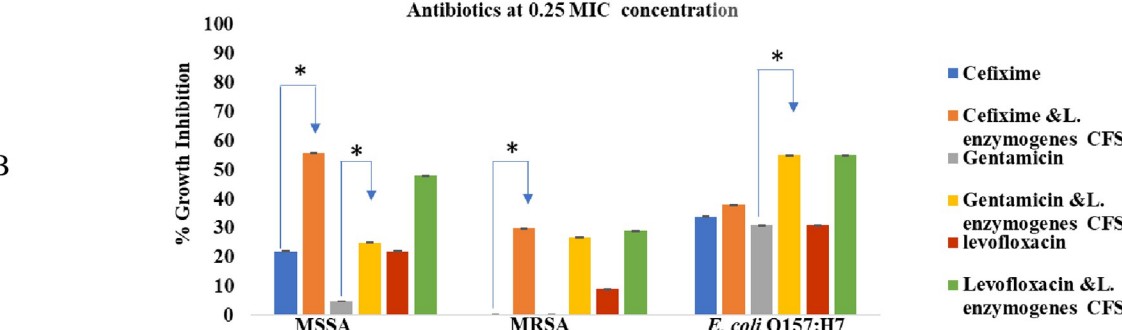

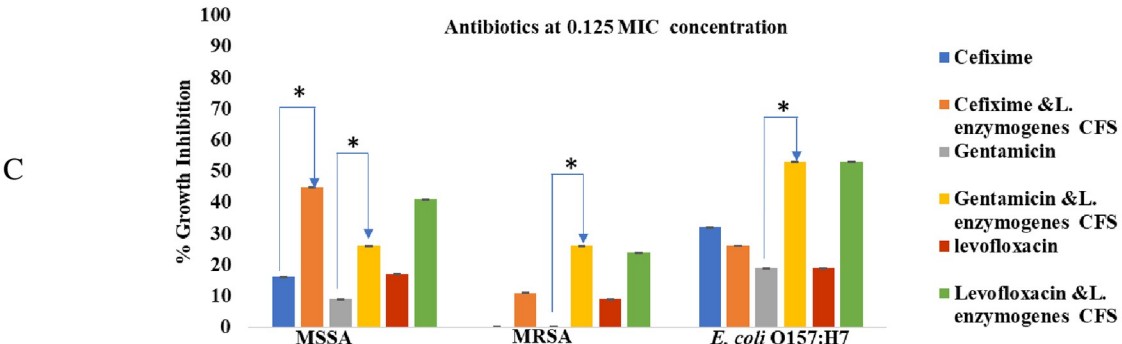

**Fig 4. *L. enzymogenes* CFS-antibiotics modulating effect against the tested pathogens.** * Statistical significance *p* < 0.005.

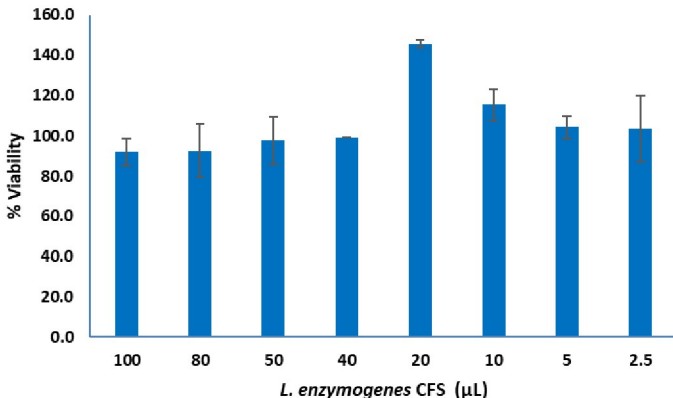

**Fig 5. Viability of human normal skin fibroblast (CCD-1064SK) at different concentrations (1.25, 2.5, 5, 10, 20, 25, 40, and 50% (v/v)) of *L. enzymogenes* CFS as assessed by MTT assay.**

combination of outer-membrane acting peptides (natural or synthetic) with antibiotics inhibiting cell-wall synthesis as a new pathway for finding effective therapy [54, 55].

*L. enzymogenes*, an environment-friendly soil-borne pathogen produces several proteolytic enzymes. Some are secreted into the culture medium while others are localized in the cell-envelope [15, 18]. In the current study, maximum proteolytic activity for *L. enzymogenes* was obtained following PBC incubation for 11 days, during which the intracellular peptidases were released into the media following cell destruction (Fig 1). *L. enzymogenes* PBC PA (4% (v/v) superior proteolytic activity (1.3 folds) compared to PB (8% (v/v), could be attributed to the higher bacterial surface area to volume ratio, hence improving dissolved oxygen and nutrients' consumption (Fig 1A).

*L. enzymogenes* CFS growth inhibitory potential was influenced by the strain and the concentration of the pathogen under investigation (Fig 3). *L. enzymogenes* CFS displayed higher inhibitory activity against *S. aureus* (MSSA and MRSA) compared to *E. coli* O157:H7 at the infectious tested bacterial concentration (Fig 3). This might be attributed to the core difference between Gram-positive and Gram-negative bacteria cell wall structure and composition [56]; *S. aureus* lack the outer lipopolysaccharide membrane enabling *L. enzymogenes* proteinases to perturb the multilayered peptidoglycan membrane, whilst in *E. coli*, *L. enzymogenes* proteinases have to cross the lipopolysaccharide layer in order to invade the monolayer peptidoglycan. As reported, *L. enzymogenes* α- Lytic protease specifically cleaves peptide bonds near small and hydrophilic amino acids as alanine, serine, threonine and valine, while β-lytic proteases specifically targets glycine, and also, the D-Ala-X bonds in bacterial cell wall peptidoglycan moieties [57]. Consequently, the higher bacteriolytic activity of *L. enzymogenes* CFS observed (Fig 3) toward *S. aureus* compared to *E. coli* O157:H7 could be attributed to its lower extent to disrupt the peptide-bridge cross-linking in *E. coli* O157:H7 peptidoglycan layer.

The decline in *L. enzymogenes* CFS antibacterial activity against higher bacterial concentration (Fig 3) could be related to the reduction in proteinases' concentration in the culture supernatant due to the binding of these proteinases with viable, lysed bacterial structures and chemical components. Thus, *L. enzymogenes* CFS percentage growth inhibitory activity is proportional to the amount of culture supernatant proteases available to each bacterium at the time of exposure.

Antimicrobials with different bacterial targets including cefixime (cell wall synthesis inhibitor) [58], levofloxacin (DNA gyrase blocker) [59] and gentamicin (protein synthesis inhibitor) [60] at sub-MIC levels in combination with *L. enzymogenes* CFS, is promising (Fig 5).

Gentamicin and levofloxacin antibacterial activity at half, quarter and one eighth the MIC against the tested pathogens has been potentiated when combined with *L. enzymogenes* CFS (Fig 5). This cooperative effect might be related to the CFS proteolytic activity toward peptido-glycan moieties, hence facilitating the accessibility of gentamicin and levofloxacin to their intracellular targets.

The reported resistance of MRSA to cefixime [61] was interestingly overcome when combined with *L. enzymogenes* CFS. Herein, cefixime-*L. enzymogenes* CFS combination may be recommended as a new strategy to combat infectious diseases caused by β- Lactam resistant MRSA.

Microbial proteases, though essentially indispensable to the maintenance and survival of their host, can be potentially damaging when present in other hosts. Antimicrobial peptides are often limited by their cytotoxicity [62]. The *in vitro* cytotoxicity is quantitatively evaluated by MTT assay. This bioassay is based on the intracellular reduction of methyltetrazolium salt by the viable cells [63, 64]. According to the International Organization for Standardization (ISO) 10993–5 [65], tested materials that reveal a reduction in cell viability by more than 30% are regarded as cytotoxic. The present research indicated high cell viability above 90% for all *L. enzymogenes* CFS tested concentrations. This finding supports the use of *L. enzymogenes* CFS with antibiotics as an adjuvant to treat bacterial infections. This combination presents a great potential for becoming a future strategy to achieve therapeutic goals. Yet, further research is required to move this therapy forward.

## Conclusion

In summary, *L. enzymogenes* CFS and antibiotics combinations showed positive antibacterial activity against the Gram-positive *S. aureus* (MRSA and MSSA) and the Gram-negative patho-gen *E. coli* O157:H7 as a new trend to combat bacterial resistance. *L. enzymogenes* CFS is a good potentiator for gentamicin and levofloxacin antibacterial activity, thus lowering the doses administered and hence reducing their side effects. In addition, cefixime's antibacterial spectrum against MRSA was recovered when combined with *L. enzymogenes* CFS. MTT assay revealed that *L. enzymogenes* CFS exhibited no significant reduction in cell viability against human normal skin fibroblast (CCD-1064SK). Given the promising results obtained so far, the appeal of using combinatory therapy will have great potential. It could represent the beginning of a modern and efficient era in the battle against multidrug resistant pathogens.

## Supporting information

**S1 Fig. Different pools of *L. enzymogenes* PBC incubated at different time intervals (Days). I: Pool A (PA, 4% (v/v)); II pool B (PB, 8% (v/v)).**
(DOCX)

**S2 Fig. Sigma's non-specific protease activity assay. I: *L*-tyrosine standard curve; II: *L*-tyro-sine standard stock solutions.**
(DOCX)

## Acknowledgments

We would like to thank participants who offered their time to participate in the survey.

## Author Contributions

**Conceptualization:** Ghadeer A. R. Y. Suaifan, Mayadah B. Shehadeh.

**Data curation:** Diana M. A. Abdel Rahman, Mayadah B. Shehadeh.

**Formal analysis:** Ala' M. Abu-Odeh.

**Funding acquisition:** Ghadeer A. R. Y. Suaifan.

**Investigation:** Ghadeer A. R. Y. Suaifan, Mayadah B. Shehadeh, Rula M. Darwish.

**Methodology:** Ghadeer A. R. Y. Suaifan, Diana M. A. Abdel Rahman.

**Project administration:** Ghadeer A. R. Y. Suaifan.

**Supervision:** Ghadeer A. R. Y. Suaifan.

**Writing – original draft:** Diana M. A. Abdel Rahman, Fahid Abu Jbara, Mayadah B. Shehadeh, Rula M. Darwish.

**Writing – review & editing:** Fahid Abu Jbara, Mayadah B. Shehadeh.

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
