## [Editor Report · Decision Letter 0]

1 Dec 2022

PONE-D-22-27664Antibiotic – Lysobacter enzymogenes proteases combination as a novel virulence attenuating therapyPLOS ONE

Dear Dr. Suaifan,

Thank you for submitting your manuscript to PLOS ONE. After careful consideration, we feel that it has merit but does not fully meet PLOS ONE’s publication criteria as it currently stands. Therefore, we invite you to submit a revised version of the manuscript that addresses the points raised during the review process.

We look forward to receiving your revised manuscript.

Kind regards,

Brahma Nand Singh

Academic Editor

PLOS ONE

Journal Requirements:

"The authors (GS and MS) would like to acknowledge the Deanship of the Scientific Research at The University of Jordan (2073, 2460) and Hamdi Mango Center for Scientist Research (2017) for financial support. The funders had no role in study design, data collection and analysis, decision to publish, or preparation of the manuscript."

"The funders had no role in study design, data collection and analysis, decision to publish, or preparation of the manuscript"

4. Please include a copy of Table 2 which you refer to in your text on Response to Reviewers file.

Additional Editor Comments (if provided):

The submitted MS has been revised accordingly. However, I have gone through the MS and found that some important earlies reports have not cited in the introduction and discussion sections of the paper. Therefore, Authors may cite the following references

RSC Adv., 2021, 11, 36459-36482; Microbial Pathogenesis (2022), 170, 105678; Bioorganic & Medicinal Chemistry Letters 27(18), 2017, 4221-4228; Trends in Biotechnology (2022); 40, 1004-1017; ACS Applied Nano Materials (2021); 4(2), 1512–1528; Scientific Reports, (2019); 9 6520; RSC Advances (2015), 5 (87), 71060-71070.
---

## [Author Response · Author response to Decision Letter 0]

29 Dec 2022

Dear Editor,

Thanks for your letter sent on Dec 1th, 2022 regarding our manuscript Ref. No.: PONE-D-22-27664, entitled: " Antibiotic – Lysobacter enzymogenes proteases combination as a novel virulence attenuating therapy". Please consider our revised manuscript entitled: which has been amended based on reviewer comments to meet PLOS ONE criteria for publication. 

Response: Thanks for the comment. We can ensure that revised version meets with PLOS ONE's style requirements. 

Response: 

Thanks for the comment. Funding Information and Financial disclosure sections in the current resubmission has been checked.

"The authors (GS and MS) would like to acknowledge the Deanship of the Scientific Research at The University of Jordan (2073, 2460) and Hamdi Mango Center for Scientist Research (2017) for financial support. The funders had no role in study design, data collection and analysis, decision to publish, or preparation of the manuscript."

"The funders had no role in study design, data collection and analysis, decision to publish, or preparation of the manuscript"

Response:

1) Funding-related text has been removed from the Acknowledgment section in the revised version.

2) Funding Statement " This research was supported by a grant (2073, 2460) from the Deanship of the Scientific Research at The University of Jordan and grant (2017) from Hamdi Mango Center for Scientist Research. The funders had no role in study design, data collection and analysis, decision to publish, or preparation of the manuscript. "

4. Please include a copy of Table 2 which you refer to in your text on Response to Reviewers file.

Response:

Apology as the article contains only table 1. There is no Table 2.

Response:

Thanks for the comment. References were revised. 

1) Reference number 7 in the submitted manuscript (Thesis) has been substituted by another article (Xu S, Zhang Z, Xie X, Shi Y, Chai A, Fan T, et al. Comparative genomics provides insights into the potential biocontrol mechanism of two Lysobacter enzymogenes strains with distinct antagonistic activities. Front Microbiol. 2022;13:966986.) 

2) New references were added (Highlighted in red in the revised manuscript) to support the use of adjuvant therapy as a tool to unlock bacterial resistance.

Additional Editor Comments (if provided):

The submitted MS has been revised accordingly. However, I have gone through the MS and found that some important earlies reports have not cited in the introduction and discussion sections of the paper. Therefore, Authors may cite the following references

RSC Adv., 2021, 11, 36459-36482; Microbial Pathogenesis (2022), 170, 105678; Bioorganic & Medicinal Chemistry Letters 27(18), 2017, 4221-4228; Trends in Biotechnology (2022); 40, 1004-1017; ACS Applied Nano Materials (2021); 4(2), 1512–1528; Scientific Reports, (2019); 9 6520; RSC Advances (2015), 5 (87), 71060-71070.

Response:

Thanks for the editor comments. References below were added to the revised version (Disscussion section) to highlight the increased interest in adjuvants therapy to unlock bacterial resistance by different application involving natural substances and nanotechnology. 

 (51. González-Bello C. Antibiotic adjuvants – A strategy to unlock bacterial resistance to antibiotics. Bioorganic & Medicinal Chemistry Letters. 2017;27(18):4221-8. 

 52. Prateeksha P, Bajpai R, Rao CV, Upreti DK, Barik SK, Singh BN. Chrysophanol-Functionalized Silver Nanoparticles for Anti-Adhesive and Anti-Biofouling Coatings to Prevent Urinary Catheter-Associated Infections. ACS Applied Nano Materials. 2021;4(2):1512-28. 

 53. Abd El-Aleam RH, George RF, Georgey HH, Abdel-Rahman HM. Bacterial virulence factors: a target for heterocyclic compounds to combat bacterial resistance. RSC Adv. 2021;11(58):36459-82.)

---

## [Editor Report · Decision Letter 1]

20 Feb 2023

Antibiotic – Lysobacter enzymogenes proteases combination as a novel virulence attenuating therapy

PONE-D-22-27664R1

Dear Dr. Suaifan,

We’re pleased to inform you that your manuscript has been judged scientifically suitable for publication and will be formally accepted for publication once it meets all outstanding technical requirements.

Kind regards,

Brahma Nand Singh

Academic Editor

PLOS ONE
---

## [Editor Report · Acceptance letter]

1 Mar 2023

PONE-D-22-27664R1 

Antibiotic – *Lysobacter enzymogenes* proteases combination as a novel virulence attenuating therapy 

Dear Dr. Suaifan:

I'm pleased to inform you that your manuscript has been deemed suitable for publication in PLOS ONE. Congratulations! Your manuscript is now with our production department. 

Kind regards, 

on behalf of

Dr. Brahma Nand Singh 

Academic Editor

PLOS ONE